# Lipid Digestibility and Polyphenols Bioaccessibility of Oil-in-Water Emulsions Containing Avocado Peel and Seed Extracts as Affected by the Presence of Low Methoxyl Pectin

**DOI:** 10.3390/foods10092193

**Published:** 2021-09-16

**Authors:** Gustavo R. Velderrain-Rodríguez, Laura Salvia-Trujillo, Olga Martín-Belloso

**Affiliations:** Department of Food Technology, University of Lleida—Agrotecnio Center, 25198 Lleida, Spain; pvvelderrain@tecal.udl.cat (G.R.V.-R.); laura.salvia@udl.cat (L.S.-T.)

**Keywords:** food-grade emulsions, polyphenols, flavonoids, agroindustrial waste, avocado by-products, functional foods

## Abstract

In this study, the digestibility of oil-in-water (O/W) emulsions using low methoxyl pectin (LMP) as surfactant and in combination with avocado peel (AP) or seed (AS) extracts was assessed, in terms of its free fatty acid (FFA) release and the phenolic compound (PC) bioaccessibility. With this purpose, AP and AS were characterized by UPLC-ESI-MS/MS before their incorporation into O/W emulsions stabilized using LMP. In that sense, AP extract had a higher content of PCs (6836.32 ± 64.66 mg/100 g of extract) compared to AS extract (1514.62 ± 578.33 mg/100 g of extract). Both extracts enhanced LMP’s emulsifying properties, leading to narrower distributions and smaller particle sizes compared to those without extracts. Similarly, when both LMP and the extracts were present in the emulsions the FFA release significantly increased. Regarding bioaccessibility, the PCs from the AS extracts had a higher bioaccessibility than those from the AP extracts, regardless of the presence of LMP. However, the presence of LMP reduced the bioaccessibility of flavonoids from emulsions containing either AP or AS extracts. These results provide new insights regarding the use of PC extracts from avocado peel and seed residues, and the effect of LMP on emulsion digestibility, and its influence on flavonoids bioaccessibility.

## 1. Introduction

Avocado industrial residues are considered an important source of bioactive compounds, such as carotenoids, tocopherols, and phenolic compounds (PC), being the latter the most abundant [1]. These residues are comprised mainly of the peel and seed of this fruit, which represents 11% and 16% of its total weight, respectively. The PC composition of both avocado residues is comprised mainly of phenolic acids (hydroxycinnamic and hydroxybenzoic acids) and flavonoids (flavonols monomers and proanthocyanidins), which are the most abundant [2,3]. Moreover, the concentration of these PCs in avocado residues is far higher than that observed in its pulp. In this regard, a diet rich in flavonoids and phenolic acids may be beneficial for human health as they are related to several protective effects like antimicrobial, anticancer, anti-inflammatory, anti-mutagenic, among others [4,5]. In previous work, the PCs from avocado residues extracts have shown antiproliferative properties against undifferentiated colon cells (Caco-2) and could be considered as functional foods functional ingredients due to their nutritional value [6]. Unfortunately, the health-related benefits of PCs depend greatly on their bioaccessibility and bioavailability, which play a critical role in their gastrointestinal and systemic functions.

The term bioaccessibility refers to the fraction of the ingested component that is released from its food matrix and solubilized in the intestinal fluids during the gastrointestinal tract to become available for intestinal absorption. [7]. In turn, the term bioavailability refers to the absorbed fraction of PC, which passed from the surface of the intestinal epithelium into the bloodstream. However, as there are no specific receptors for PCs in the intestinal epithelium, their bioavailability depends on the bioaccessibility of PCs and their passing through the intestinal mucus layer and the underlying epithelium [8]. In that sense, as the molecular weight and lipophilicity are related to bioaccessibility, it can be considered that the bioaccessible fraction of PCs is comprised of those with molecular weight and lipophilicity that allows their solubility in the intestinal fluids. Therefore, the bioaccessibility of PCs has even greater relevance than its concentration in the food products where it is located. Thus, there is a growing need for developing new delivery systems that help in dispersing, protecting, carrying, and increasing the bioaccessibility of PC.

Emulsions are mainly biphasic dispersions with either water dispersed in oil (W/O) or oil dispersed in water (O/W) stabilized by a surfactant, the latter being the most commonly used for traditional food products [9]. The use of this system has revealed promising results as carriers of bioactive compounds, increasing their bioavailability, and allowing better penetration of the mucous layer [10,11]. In this regard, the use of biopolymers, such as proteins and polysaccharides, as surfactants is gaining interest as they are considered natural, safe, and sustainable ingredients able to successfully stabilize O/W emulsions [12]. In terms of emulsion stability, pectin has emulsifying properties that are largely owned to its ability to increase the apparent viscosity of the continuous phase in emulsions [13]. In addition, it has been reported that pectin form an interpolymeric gel network around the emulsion’s oil droplets, which provides electrostatic and steric repulsions that helps to prevent droplets from flocculation and coalescence [12]. Moreover, it has also been suggested that PCs may influence the conformational flexibility of these pectin interpolymeric networks by forming complexes that improve the physical stability of oil-in-water (O/W) emulsions [14]. Recently, Velderrain-Rodríguez, Salvia-Trujillo, González-Aguilar, and Martín-Belloso [1] reported that the emulsifying activity of low methoxyl pectin (LMP) is enhanced by the presence of avocado peel and seed extracts, leading to long-term stable O/W emulsions with a reduced formation of secondary oxidation products. Moreover, these emulsions containing PCs from avocado residues, and stabilized using LMP, had a higher colloidal and oxidative stability compared to nanoemulsions stabilized with Tween 80. These authors suggested that this stability enhancement may be related to some type of interaction or the formation of complexes between the PCs from avocado peel and seed extracts and the LMP.

However, plant cell wall polysaccharides, such as LMP, are generally resistant to digestion in the upper gastrointestinal tract. Thus, when LMP is used as a surfactant, the changes occurring in the emulsion microstructure during digestion may be influenced by the LMP resistance affecting the lipids digestibility and, thus, the bioaccessibility of some bioactive compounds [15,16]. In the presence of pectin, a decreased rate and extent of lipid digestion in O/W emulsions have been reported, and the effect of pectin on the emulsions’ digestibility has been attributed to several mechanisms. For example, lipases might have limited access to anchor at the oil/water interfaces possibly due to (i) an increased viscosity of the aqueous medium, (ii) droplets flocculation, or (iii) electrostatic interactions between complexes formed around the oil droplets might also have some influence on either lipid digestibility or PC bioaccessibility during the different digestion steps [4]. Nevertheless, there are still scarce studies addressing the impact of LMP in the digestibility of O/W emulsions and the subsequent PC bioaccessibility using these delivery systems.

The natural occurrence of LMP in agro-industrial residues, along with its health-related properties and its ability to produce long-term stable emulsions when combined with PC, highlights its importance as a natural emulsifier to be used in novel food products based on emulsions. Thus, the aim of this study was to evaluate the digestibility of O/W emulsions using LMP as surfactant and in combination with avocado peel (AP) or seed (AS) extracts, in terms of its lipid digestibility and PC bioaccessibility. Firstly, in order to identify and quantify the main PCs present within the AP and AS extracts, identification and quantification of their individual PCs were performed using a UPLC-ESI-MS/MS. Secondly, the effect of LMP on the formation of O/W emulsions containing AP or AS extracts was evaluated in terms of its volume mean diameter (d_4,3_) and particle size distribution using a static light scattering technique. Lastly, the effect of LMP on the lipid digestibility and PC bioaccessibility was measured in terms of the free fatty acid (FFA) release and the PC content (quantified by UPLC-ESI-MS/MS) after simulated digestion conditions.

## 2. Materials and Methods

‘Hass’ avocado fruit and avocado oil (ETHNOS™) were purchased from a local market in Lleida, Spain. LMP was kindly donated by Herbstreith & Fox KG, Neuenbürg, Germany. The standards hydroxytyrosol, tyrosol, quercetin-3-O-glucoside, quercetin-3-O-rutinoside (rutin), kaempferol-3-O-glucoside, isorhamnetin, dimer B2 were purchased from Extrasynthese (Genay, France), whereas p-hydroxybenzoic acid, 3,4-dihydroxybenzoic acid (protocatechuic acid), p-coumaric acid, caffeic acid, ferulic acid, chlorogenic acid (5-O-CQA), catechin, epicatechin were from Sigma-Aldrich (St. Louis, MO, USA). The digestive enzymes (pepsin and pancreatin from porcine pancreas) and bovine bile were purchased from Sigma-Aldrich, Inc. (St Louis, MO, USA). Ethanol and other solvents were purchased from Fischer Scientific (Leicestershire, UK) and Scharlau S.L. (Barcelona, Spain). The ultrapure water used to prepare all solutions of this study was obtained using a Milli-Q filtration system (18.2 mΩ, Merck Millipore, Madrid, Spain).

### 2.1. Avocado Peel and Seed Extracts

The avocado peel (AP) and seed (AS) extracts were obtained from the peels and seeds, previously separated from the washed avocado fruit, as described by Velderrain-Rodríguez, Salvia-Trujillo, González-Aguilar, and Martín-Belloso [1]. Briefly, dried AP and AS were macerated with an 80% ethanol solution, incubated for 20 h at 40 °C using an orbital shaker, and subsequently centrifuged at 5000 rpm for 10 min at 4 °C and filtered. Then, the ethanol was removed using a rotary evaporator under vacuum at 40 °C and subsequently lyophilized using a laboratory freeze-drier (Telstar Cryodos, Spain), and powdered using a kitchen blender.

### 2.2. Identification and Tentative Quantification of Individual Phenolic Compounds

Phenolic compounds (PC) and derived metabolites in avocado peel and seed extracts, were identified and quantified as described by López-Gámez et al. [7], with minor modifications. AcQuity Ultra-Performance™ liquid chromatography (UPLC), coupled to a triple quadrupole detector (TQD) mass spectrometer (all from Waters, Milford, MA, USA) was used. The analytical column was an AcQuity BEH C18 column (100 mm × 2.1 mm i.d., 1.7 μm,) equipped with a VanGuard™ Pre-Column AcQuity BEH C18 (2.1 × 5 mm, 1.7 μm), also from Waters. During the analysis, the column was kept at 30 °C, and the flow rate was 0.3 mL min^−1^. Mobile phases were acetic acid (0.2%) and acetonitrile, and elution gradients are shown in Table 1. Tandem MS analyses were carried out on a triple quadrupole detector (TQD) mass spectrometer (Waters, Milford, MA, USA) equipped with a Z-spray electrospray interface (ESI). Ionization was achieved using the ESI operating in the negative mode [M−H]^−^ and the data were acquired through selected reaction monitoring (SRM). Two SRM transitions were selected, the most sensitive one was used for quantification, and the second for confirmation purposes. The dwell time established for each transition was 30 ms. The SRM transitions, cone voltages, and collision energies for each compound were collected. Data acquisition was carried out with the MassLynx 4.1 software (Waters, Milford, MA, USA). Results were expressed on a dry weight basis (mg per 100 g of extract).

### 2.3. Emulsion’s Formation

The O/W emulsions, using LMP as the surfactant, was prepared as described by Velderrain-Rodríguez, Salvia-Trujillo, González-Aguilar, and Martín-Belloso [1], with minor modifications. The O/W emulsions were formulated with avocado oil (10% *w*/*w*), 10 mM citric acid buffer (88.5% *w*/*w*), and LMP (1% *w*/*w*). In those emulsions containing AP or AS extracts, the extracts were directly blended with the coarse emulsion without prior solubilization. For this, 0.5% (*w*/*w*) of AP or AS extracts were added to the former mixture of components and homogenized at room temperature (20–25 °C) using an Ultra-Turrax T25 (IKA-Werke, Staufen, Germany) at 15,000 rpm for 3 min. Emulsions were obtained after passing coarse emulsions three times through a microfluidizer (model M110-P, Microfluidics, Newton, MA) working at 7000 psi. The microfluidizer was also coupled to a cooling water bath to prevent an increase in temperature and maintain it between 20–25 °C. Emulsions without extracts were also homogenized following the same protocol and subsequently referred to as “blank emulsions”. Moreover, oil/water dispersions containing or not the AP and AS extracts, but without pectin, were also prepared following the exact same protocol for comparison purposes of the emulsion’s digestibility. These will be subsequently referred to as “blank dispersions”, “dispersions with AP” or “dispersions with AS extracts”. As the dispersions did not contain pectin or any other surfactant, these were highly unstable and prepared to be used immediately.

### 2.4. Emulsion Droplet Size and Droplet Size Distribution

The particle size and particle size distribution were measured by static light scattering technique using a Mastersizer 3000 (Malvern Instruments Ltd., Worcestershire, UK). Distilled water was used as the dispersant phase, with a refractive index of 1.465 for avocado oil. Data were reported as volume-weighted averages (d_4,3_), based on the Mie Scattering theory, and expressed in µm.

### 2.5. Emulsion In Vitro Digestibility

The in vitro digestibility of emulsions (with pectin) or dispersions (without pectin) containing AP and AS extracts were performed according to Minekus et al. [17], with minor modifications, consisting of a gastric and intestinal phase. Simulated gastric fluid solution (SGF) was composed of a mixture of electrolytes [0.5 M KCl, 0.5 M KH_2_PO_4_, 1 M NaCl, 2 M NaCl, 0.15 M MgCl_2_(H_2_O), 1 M (NH_4_)_2_CO_3_] dissolved in 20 mL of milli-Q water. Then, 18.2 mL of SGF was acidified adding 1.8 mL of 0.02 M HCl and used to dissolve pepsin (8.8 mL/mL). Gastric digestion was performed by mixing 20 mL of emulsion with 20 mL of SGF solution. The mixture was incubated under subdued light conditions for 2 h at 37 °C and continuous agitation using an orbital shaker working at 100 rpm. Afterward, 30 mL of the chyme was placed in a water bath at 37 °C. Finally, 3.5 mL of bile salts (54 mL/mL) and 1.5 mL of 0.1 M calcium chloride solutions were added to the digested emulsions, adjusting pH to 7.0 prior addition of 2.5 mL of pancreatin (75 mL/mL) solution. The emulsion digestibility test during intestinal digestion was performed using pH-stat equipment (Metrohm USA Inc., Riverview, FL, USA), as reported by Gasa-Falcon et al. [15]. The free fatty acid (FFA) release in percentage was calculated according to Equation (1):(1)$$FFA\;\left(\%\right)=\frac{V_{NaOH}\times \;C_{NaOH}\times M_{oil}}{2\times m_{oil}}\times 100$$
where *V_NaOH_* represents sodium hydroxide volume needed to neutralize free fatty acids released during intestinal digestion, whereas *C_NaOH_* is the Molar concentration of sodium hydroxide (0.25 M), *M_oil_* is the molecular weight of corn oil (800 g/mol) and *m_oil_* is the total oil weight within emulsions.

### 2.6. Bioaccessibility of Phenolic Compounds

The remaining digests, which contained the released PC, was centrifuged (Avanti J-26 XP, Beckman Coulter, Pasadena, CA, USA) at 5000 g for 15 min at 4 °C. The supernatant was filtered through PTFE 0.45 μm filters to obtain the water-soluble fraction where the PCs were contained. The quantification was performed using UPLC-MS as described previously, to subsequently obtain the sum of the individual PCs and expressed the results as the bioaccessibility of phenolic acids, flavonoids, or terpenes. Therefore, the bioaccessibility of PCs was calculated according to Equation (2):(2)$$Bioaccesibility\;\left(\%\right)\;=\;\frac{CC_{digested}}{CC_{undigested}}\;\times \;100$$
where *CC_digested_* corresponds to the overall concentration of the different PC groups quantified in the digested fraction after centrifugation and filtration, whereas the *CC_undigested_* is the concentration of PCs in the undigested fraction.

### 2.7. Statistical Analysis

All the experiments were carried out in duplicate, with at least three measurements by each assay performed. Statistical differences were analyzed by one-way ANOVA and a Tukey–Kramer multiple comparison test (*p* < 0.05) using the statistical software NCSS 2007 (NCSS, Kaysville, UT, USA).

## 3. Results and Discussions

### 3.1. Tentative Quantification of the Individual Phenolic Compounds in Avocado Peel and Seed Extracts by UPLC-ESI-MS/MS

The quantification of the individual phenolic compounds (PC) in avocado peel (AP) and seed (AS) extracts is shown in Table 2. An accurate quantification was performed for those PCs with a reference standard, while the rest were tentatively quantified with the available reference standards of their respective PC type (hydroxybenzoic acids, hydroxycinnamic acids, flavonols monomers, proanthocyanidins, and terpenes). Both accurate and tentative quantifications were solely considered a semi-quantitative description of PC content in the AP and AS extracts. Moreover, for comparison purposes, the total content for the different groups (phenolic acids, flavonoids, and terpenes) and the total PCs in AP and AS extracts were calculated as the sum of the individual PC concentrations. In this study, 77 different compounds were identified and quantified in the extracts obtained from avocado residues, of which 69 different compounds were found in AP extract and 57 compounds in AS extract. According to these results, the AP extract had the highest content of total PC, as it had a 4.5-fold content (6836.32 ± 64.66 mg/100 g of extract) of total PCs compared to that observed in AS extract (1514.62 ± 578.33 mg/100 g of extract). Regarding the different PC groups, the content of phenolic acids in AP extract was about 1111.54 ± 11.25 mg/100 g of extract, 2.9 times higher than the phenolic acid content in AS extract (377.98 ± 111.64 mg/100 g of extract). Moreover, 5-fold content of flavonoids was observed in AP (5721.88 ± 51.73 mg/100 g of extract) compared to that in AS (1135.77 ± 456.57 mg/100 g of extract). The terpene content in AP extracts followed a similar trend, as it had a 4.5-fold content (2.82 ± 1.86 mg PC/100 g of extract) compared to AS extracts (0.62 ± 0.13 mg PC/100 g of extract).

In this study, flavonoids were the most representative group of PCs in both AP and AS extracts. These results have shown that 37 different types of flavonoids were found in AP extracts, being the procyanidin dimer type B and epicatechin those of higher concentration, with 2262.0 ± 63.00 mg/100 g of extract and 1891.00 ± 75.70 mg/100 g of extract, respectively. As for the flavonoids in AS extract, 31 different types were found, of which over 95% of the total content was comprised of epicatechin, catechin, procyanidin trimer (type A), and procyanidin dimer (type B). Among phenolic acids, 31 different compounds were found in AP, and about 87.19% (969.20 ± 9.21 mg/100 g of extract) corresponded to 5-*O*-caffeoylquinic acid, followed by protocatechuic acid glucoside, which represented the 2.34% (26.10 ± 0.27 mg/100 g of extract) of the total phenolic acid content. For AS extracts, 25 compounds were found, with 3-O-caffeoylquinic acid and salidroside being more than 85% of its total phenolic acid content. Regarding the terpene content, penstemide was the only compound found in both AP and AS extracts.

Among flavonoids, the procyanidins are monomeric and oligomeric forms exclusively of (epi)catechin units linked together, which have been related to several health benefits [18,19]. Thus, procyanidins, as bioactive food compounds, exert physiological and cellular activities that promote homeostasis [20]. The two main types of procyanidin oligomers found in plant-based foods are type A and type B procyanidin oligomers. On the one hand, type A procyanidins oligomers have two linkages, which include a C4–C8 bond and an additional ether bond, and their trimers and tetramer have molecular masses of 864 and 1152 Da, respectively [21]. The most common sources of type A procyanidins are plums, avocados, peanuts, curry, cinnamon, and cranberries plums, avocados, peanuts, curry, cinnamon, and cranberries [22]. On the other hand, the type B procyanidin oligomers contain flavan-3-ol units that singly link through C4 → C8 and/or C4 → C6 bonds, and their trimers and tetramer have molecular masses of 866 and 1154 Da, respectively [23]. The main dietary sources of type B procyanidins are grapes, blueberries, cocoa, sorghum, and apples [22].

In a similar study, Figueroa, Borrás-Linares, Lozano-Sánchez, and Segura-Carretero [3] found a total of 61 PCs in AP extracts from the ‘Hass’ variety using a HPLC-DAD-ESI-QTOF-MS. In agreement with the results of the present work, these authors reported that the most representative groups of PCs found in AP were flavonoids (Procyanidins, flavonols) and phenolic acids (hydroxybenzoic and hydroxycinnamic acids). Similarly, Figueroa, Borrás-Linares, Lozano-Sánchez, and Segura-Carretero [2] reported a higher number of PCs in AS, using the same avocado variety, as they were able to detect 75 different compounds using HPLC-DAD-ESI-QTOF-MS. These authors reported that polymeric procyanidins (condensed tannins) were the most representative group of PCs in AS, as they found 29 different compounds, with procyanidin type A content in a higher proportion than that found for type B. Interestingly, according to these studies, the seed is suggested as a major source of PCs compared to peel, which can be related to the different extraction process performed by these authors. Conversely, different studies have reported a higher content of bioactive compounds in AP, compared to pulp and AS, either by the colorimetric or HPLC-DAD methods [1,24,25]. In these studies, catechin and epicatechin are suggested as the main flavonoids found in AP and AS extracts. Nevertheless, there’s a lack of studies regarding the stability of these extracts from avocado residues during conventional food processing conditions.

### 3.2. Effect of Avocado Peel and Seed Extracts on the Formation of O/W Emulsions Using LMP as Surfactant

The effect of LMP on the particle size distribution of O/W emulsions containing AP or AS extracts is shown in Figure 1. In this study, only the emulsion’s formation is addressed, as the colloidal and oxidative stability of these emulsions using LMP as a surfactant (with and without AP or AS extracts) was described in a previous work by Velderrain-Rodríguez, Salvia-Trujillo, González-Aguilar, and Martín-Belloso [1]. These authors also reported that AP and AS extracts contain PCs with interfacial activity that can enhance the formation of O/W emulsions. Thus, dispersions without any surfactant were also obtained, with or without AP and AS extracts, only for comparison purposes. As shown in Figure 1A, the dispersions containing AP or AS extracts can form emulsions without using LMP as a surfactant. However, these dispersions led to emulsions with bimodal particle size distributions. In that sense, even when a higher volume of particles between 0.21 and 3.55 µm was obtained in the presence of AS extract, the presence of larger particles (between 4.03 and 111.47 µm) also occurred. As for those with AP extract, the higher volume of particles was between 3.52 and 40.14 µm, followed by smaller populations between 0.23 and 3.52 and another between 4.32 and 23.28 µm. The blank dispersions did not form emulsions, thus, there was no particle size distribution that could be compared to that of the former.

According to the results shown in Figure 1B, all the emulsions containing LMP led to monomodal particle size distributions. The emulsions without extracts and containing LMP had a narrow particle size distribution of between 0.67 and 3.55 µm. Moreover, smaller particle sizes with narrower distributions, between 0.35 and 1.87 µm or 0.40 and 2.13 µm, were observed in the presence of AP and AS, respectively. In agreement, Velderrain-Rodríguez, Salvia-Trujillo, González-Aguilar, and Martín-Belloso [1] reported that the presence of AP and AS extracts led to O/W emulsions (using LMP as surfactant) with a smaller particle size between 1 and 4 µm. These authors suggested that the PC molecules within AP and AS extracts may be acting as co-surfactants, enhancing the emulsifying properties of LMP. In that sense, it has been reported that the emulsification properties of LMP are related to an increase of steric repulsions and/or continuous phase viscosity, or in some cases to the adsorption of protein fractions positioned at the hydrophobic cavities of pectin chains [26].

The presence of PCs in the continuous phase of O/W emulsions may enhance the emulsifying properties of LMP by promoting its interaction with the interface and the lipid phase of the emulsions [27,28,29]. Thus, to evaluate the presence of PCs either at the continuous phase or the oil-water interface, the octanol-water partition coefficient (Log P) value of PCs should be considered as it describes the lipophilicity of these molecules. In general, Log P values above zero indicates a higher solubility in octanol (lipophilic compounds), whereas negative Log P values indicate a higher solubility in water (hydrophilic compounds) [30]. In that sense, according to the in silico analysis of the AP and AS extracts performed by Velderrain-Rodríguez, Quero, Osada, Martín-Belloso, and Rodríguez-Yoldi [6], the Log P value (between 0 and 4) of some of its phenolic acids (such as p-hydroxybenzoic, vanillin, vanillic, syringic, protocatechuic, hydroxytyrosol, caffeic and ferulic acids), flavonoids (such as catechin, epicatechin, naringenin, sakuranetin, and luteolin), and the terpene penstemide suggests that these PC molecules might be able to be positioned at the oil–water interface. Moreover, the presence of large PC molecules in AP and AS extracts, such as polymeric procyanidins (condensed tannins), may enhance those steric repulsions or ease the adsorption of LMP at the oil-water interface. In addition, pectin-anthocyanins complexes may enhance the formation of O/W emulsions and may be occurring due to the intermolecular π-π stacking of anthocyanins and hydrogen bonds from hydroxyl groups in both molecules [31]. The π-π stacking is referred to the non-covalent attractive interaction that involves π-electron systems, often formed between two aromatic rings of the anthocyanins. The presence of anthocyanins was previously reported by Velderrain-Rodríguez, Salvia-Trujillo, González-Aguilar, and Martín-Belloso [1] in both AP and AS extracts, and could be related to the content of condensed tannins discussed in the previous section.

### 3.3. Effect of LMP on the Digestibility of Emulsions

The lipid digestibility of O/W emulsions containing AP and AS extracts and LMP was evaluated in terms of the changes in volume mean diameter (d_4,3_) and the particle size distribution of oil droplets (Figure 2 and Figure 3, respectively) after simulated gastric and intestinal phases. Moreover, the digestibility of the emulsions’ oil was also assessed in terms of the free fatty acid release (%) during simulated small intestine conditions (Figure 4).

#### 3.3.1. Colloidal Stability of Emulsions during Digestion

According to the results shown in Figure 2A, the volume mean diameter of the dispersions containing AP and AS extracts had no significant changes after gastric digestion conditions. However, the volume mean diameter significantly increased after intestinal digestion conditions, especially in those dispersions with AP extract. Thus, no differences were observed (in terms of its mean volume diameter) after the intestinal phase between those dispersions containing AP extract and the blank dispersion, as both had a diameter size between 130 and 140 µm. Interestingly, the dispersions with AS extracts had a smaller volume mean diameter of 55.02 ± 4.38 µm, suggesting that there are some PC species in AS extracts that could be enhancing the colloidal stability of these dispersions during simulated digestion conditions. Regarding the emulsions stabilized with LMP, the size of the volume mean diameter increased after both gastric and intestinal conditions (Figure 2B). In that sense, after gastric digestion conditions, an increase up to 31.87 ± 1.84 µm and 31.55 ± 0.55 µm was observed for emulsions containing AP and AS, respectively. Subsequently, after intestinal digestion conditions, the volume mean diameter increased up to 202.22 ± 23.25 µm and 45.98 ± 10.45 for those emulsions containing AP and AS, respectively.

The changes in the particle size distribution of oil droplets occurring through the gastric and intestinal digestion conditions are displayed in Figure 3. As shown in Figure 3A, compared to blank dispersions, those containing the AP and AS extracts had a higher volume of particles between 4.58 ± 51.82 µm and 1.87 ± 40.14 µm, respectively, after gastric digestion conditions. For those dispersions containing AP extracts, populations of a particle size between 100 and 1000 µm occurred after intestinal digestion (Figure 3B), whereas those containing AS extracts had a higher volume of particles at a smaller range (between 7.63 ± 143.89 µm). Regarding emulsions stabilized with LMP, the results in Figure 3C show that after gastric digestion, the particle size of blank emulsions was in a smaller size range (between 0.67 and 4.0 µm) compared to that observed for the emulsions containing either AP or AS extracts (between 10 and 100 µm). After intestinal digestion (Figure 3D), the higher volume of particles was between 21.20 and 185.75 µm in emulsions containing AP extracts, and between 2.42 and 40.14 µm in emulsions containing AS extracts. Thus, the occurrence of large particle size populations may explain the results observed for the volume mean diameter of emulsions, as this is highly influenced by large particles.

The blank emulsions had no significant changes in their particle size after gastric digestion, probably as the viscosity of the continuous phase made the action of lipase difficult. As suggested by Wijaya et al. [32], polysaccharides like pectin can form an interpolymeric gel network around the emulsion droplets due to electrostatic interactions that are not easily displaced. However, in the presence of AP and AS extracts weaker interpolymeric gel networks might be formed due to the occurrence of non-covalent complexes between pectin and the phenolic species from these extracts. These interactions are favored by the existence of hydrophobic cavities in the LMP chains, where those PCs of higher lipophilicity might be positioned. Thus, the acidic environment of the gastric phase might be able to affect the strength of the gel network formed by these complexes around the emulsion oil droplets. As suggested by Luo et al. [33], during the acidic conditions of the gastric phase (pH 2.5) the charge of LMP decreases and, thus, the electrostatic repulsion between the oil droplets is weaker. Therefore, the appearance of larger particle size populations after the gastric digestion of these emulsions may be attributed to the formation of weaker interpolymeric gel networks affected by the environmental conditions that changed the nature of the colloidal interactions (such as pH and ionic strength).

#### 3.3.2. Lipid Digestibility

In Figure 4, the in vitro lipid digestibility of O/W dispersions and O/W emulsions was assessed in terms of the FFA release (%) during small intestine conditions. The results shown in Figure 4A, suggest that there are differences in the FFA release during intestinal digestion conditions in the presence of either AP or AS extracts. Interestingly, a higher FFA release is observed for the dispersions containing AS extracts during intestinal digestion, with an 86.94 ± 5.09% after 120 min. However, there were no differences between the FFA release from the blank dispersions and those containing AP extracts during the first 60 min. Nevertheless, after 120 min of intestinal digestion, the blank dispersions had a higher FFA release (72.48 ± 3.14%) compared to the dispersions containing AP extract (63.15 ± 5.69%). As for the emulsions stabilized with LMP, the FFA release was enhanced in the presence of both extracts. As it is shown in Figure 4B, the emulsions containing AS extracts had an FFA release of 91.14 ± 5.09%, whereas those with AP extract had an FFA release of 81.48 ± 5.68%. Conversely, the presence of LMP had no effect on the digestibility of the oil droplets in the emulsions without extracts, as a similar FFA release (72.48 ± 2.32%) was observed. Thus, these results suggest that the complexes formed between LMP and PCs from AP and AS extracts might enhance lipid digestion during intestinal conditions.

The increased FFA release in emulsions containing LMP and the extracts might be explained in terms of the emulsion’s particle size. According to Salvia-Trujillo et al. [34], when decreasing droplet size, the lipid digestion rate (FFAs released per unit time) increases. Moreover, after the small intestinal phase, depending on the emulsion type, large oil droplets might undergo coalescence and remain as undigested oil droplets [35]. In agreement with that, the highest FFA% was observed in emulsions containing AS extracts and LMP, which were also the emulsions with the smallest particle size in this study. Thus, these results suggest that the higher and faster FFA generation rate in O/W emulsions containing AP or AS extracts can be attributed to the smaller droplet size and higher surface area for anchoring lipase as well as a thinner interfacial layer. Moreover, the use of binary complexes, such as polysaccharide-polyphenol complexes, have been verified to have better emulsification properties or special interfacial structures compared with single components to improve the emulsions’ physicochemical properties [36]. For example, the polysaccharide-polyphenol complexes formed by LMP and PCs reduces the formation of secondary oxidation products and increases the colloidal stability of O/W emulsions for over 50 days of storage [1].

#### 3.3.3. Phenolic Compounds Bioaccessibility

Lastly, the bioaccessibility of PCs from AP and AS extracts within the O/W dispersions (without LMP) and emulsions (with LMP) is shown in Figure 5. The bioaccessibility of PCs from emulsions containing AP and AS extracts was calculated using the sum of the individual phenolic acids, flavonoids, and terpenes quantified by UPLC-ESI-MS/MS. The sum and individual content of all the PC groups is shown in Appendix A. In that sense, the bioaccessibility of the dispersions without LMP is shown in Figure 5A,C. According to Figure 5A, only 38.44 ± 5.17% of the phenolic acids from the dispersions containing AP extracts were bioaccessible after intestinal digestion conditions. Once the simulated digestion finished, only 15 of the 31 individual phenolic acids found in the AP extract could be detected (Appendix A). Moreover, these results showed that flavonoid content was 2-fold, and those different terpenes (cinchonain and nudiposide) could be detected in these dispersions after digestion conditions, leading to a bioaccessibility of over 100% for both groups. As for dispersions containing AS extract, all PC groups had a bioaccessibility of over 100%, and an increased sum of the individual PCs (Appendix A) was observed after digestion conditions. Even when some individual compounds from the phenolic acids and flavonoids groups were not found after the dispersion’s digestion, the concentration of phenolic acids and flavonoids significantly increased compared to that in AS extract. As for terpenes, penstemide remained the only species found in AS extract after the dispersion’s digestion.

The differences between the bioaccessibility of phenolic acids and flavonoids observed in those dispersions containing AP extracts could be related to their different susceptibility to degradation during intestinal digestion. As reported by de Araújo et al. [37], phenolic acids are usually found in simple structures, whereas flavonoids are present mainly in their glycosidic forms linked to carbohydrates (including glucose, rhamnose, maltose, etc.), which can provide them higher stability under gastrointestinal conditions [38]. Moreover, the higher bioaccessibility of procyanidin and other flavonoids after intestinal digestion could be attributed to the hydrolysis of isomers, such as trimers, tetramers, and pentamers during gastric and intestinal digestions. Thus, the enzymes and the digestive conditions could facilitate either the release of these compounds bound to the matrix or the subsequent release of any highly polymerized or insoluble PCs present in the dispersions [39]. In brief, the increase of flavonoids during intestinal digestion and their relative stability during the gastric and intestinal phase could be linked to their greater release and chemical stability in glycosylated forms.

On the other hand, the differences between the bioaccessibility of phenolic acids from AP and AS extracts could be related to their chemical structures or the polymerized form in which they were extracted. According to Yu et al. [40], dicaffeoylquinic acid could be degraded into its free acids form (caffeic acid and quinic acid) after gastrointestinal digestion conditions. Moreover, an increased bioaccessibility may be due to the disruption of the cell walls and, thus, an easier release of phenolic compounds from the food matrix. Furthermore, an increased pH can cause the degradation of certain anthocyanins, leading to the release of phenolic acids as degradation products [41]. Therefore, the results of this study suggest that the PCs in AS extracts may be present as highly polymerized molecules or in their glycosidic forms that are degraded into the PC-free monomeric forms during the different digestion conditions, increasing the overall bioaccessibility of the PC groups.

As for the emulsions stabilized using LMP, lower bioaccessibility values were observed either in emulsions with AP or AS extracts. As shown in Figure 5B, a bioaccessibility of 49.28 ± 3.69% and 49.15 ± 3.08% for phenolic acids and flavonoids, respectively, in emulsions containing AP extracts. According to these results, the presence of LMP reduced around 51% of the bioaccessibility of flavonoids in these emulsions. As for those containing AS extracts, flavonoids had 42.29 ± 6.82%, an around 60% decrease compared to those emulsions without LMP. Regardless of the extract, these results suggest that the presence of LMP reduces the bioaccessibility of flavonoids, possibly due to some type of non-covalent pectin-flavonoid interaction. In that sense, the main flavonoids involved in those interactions could be epicatechins and procyanidins type B, as a significantly lower concentration was observed after the emulsion’s digestion (Appendix A). In agreement, Tomas et al. [42] stated that the interactions between phenolic compounds and dietary fibers are mostly driven by a combination of hydrogen bonding, van der Waals forces, electrostatic attraction, hydrophobic contact, strong covalent bonding (esterification), or physicochemical entrapment. Thus, LMP can act as an entrapping matrix thereby limiting the bioaccessibility of these flavonoids in emulsions containing AP and AS extracts.

Therefore, the bioaccessibility of PCs is highly influenced by their chemical structure, concentration in the food matrix, and interactions with other constituents. The increment of certain PC groups present in O/W emulsions after digestion conditions could be triggered by changes in their chemical structure and/or microstructural modifications that ease their release. Moreover, processing conditions may also cause structural changes in the food matrixes containing PCs, thus, affecting either positively or negatively the bioaccessibility of these compounds in food products [43]. In that sense, further studies evaluating the impact of food processing on the bioaccessibility of food products, including either AP or AS extracts, need to be performed. Furthermore, the results of this study suggest that the presence of LMP reduces the bioaccessibility of flavonoids from the AP or AS extract present in O/W emulsions, possibly due to some type of entrapment or interaction.

## 4. Conclusions

This work proved that avocado peel (AP) and seed (AS) are an important sources of phenolic compounds (PCs) and that they can be successfully incorporated into food-grade emulsions stabilized using low methoxyl pectin (LMP). The main bioactive compounds found in the AP and AS extracts added to the emulsions were flavonoids, followed by phenolic acids and terpenes. Moreover, the results of this study showed that the presence of AP and AS extracts enhanced the pectin’s emulsifying properties. As to the lipid digestibility, in combination with the extracts, LMP significantly increased the lipid digestibility of the emulsion’s oil droplets. Regarding PC bioaccessibility, the presence of non-covalent interactions with LMP or the formation of flavonoid-pectin complexes could be responsible for the reduced bioaccessibility of flavonoids. Procyanidins type B, catechin, and epicatechin could have a higher affinity to LMP and might be involved in some type of complex that limits the flavonoids bioaccessibility. Regarding the bioaccessibility of phenolic acids, compared to those from AP extract, the phenolic acids from AS extract had higher bioaccessibility, probably due to their occurrence as highly polymerized molecules or in their glycosidic forms which can provide them higher stability under gastrointestinal conditions.

Hence, this work reveals important information regarding the use of LMP as a surfactant in O/W emulsions containing phenolic compounds extracts. In that sense, the results from this study provide new insights regarding the effect of LMP on emulsions digestibility and its influence on flavonoids bioaccessibility. Future studies should be focused on studying the impact of food processing on these compounds and the type of interactions occurring between LMP and flavonoids to enhance its bioaccessibility in different functional food products.

## Figures and Tables

**Figure 1 foods-10-02193-f001:**
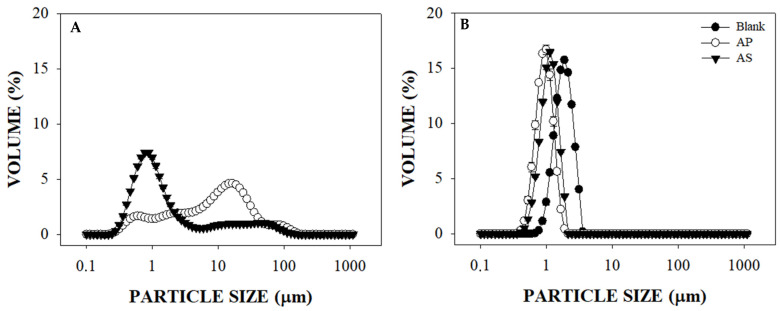
Effect of peel and seed extracts on the particle size distributions of oil-in-water (O/W) dispersions (without surfactant) and emulsions using low methoxyl pectin (LMP) as surfactant. (**A**) = O/W dispersions without LMP formed with avocado seed (AS) or avocado peel (AP) extracts; (**B**) = Emulsions containing LMP without or with AS or AP.

**Figure 2 foods-10-02193-f002:**
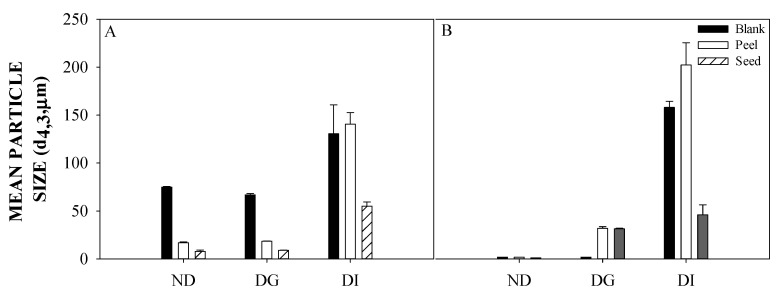
Changes in the particle size of oil-in-water (O/W) dispersions and emulsions, stabilized with low methoxyl pectin (LMP), containing avocado peel and seed extracts during simulated gastric and intestinal digestion conditions. (**A**) = O/W dispersions; (**B**) = Emulsions containing LMP. ND = Non digested; DG = Gastric digestion; DI = Intestinal digestion.

**Figure 3 foods-10-02193-f003:**
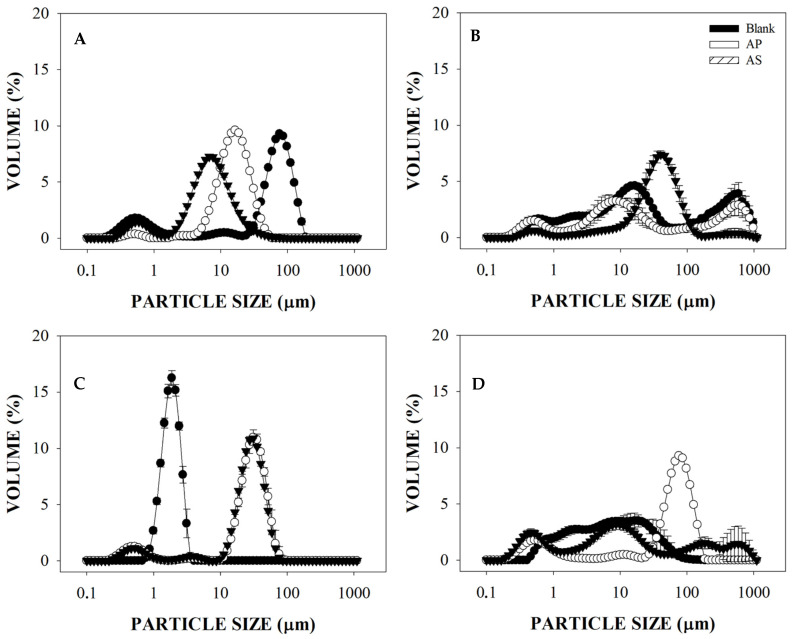
Effect of low methoxyl pectin (LMP) on the particle size distribution of oil-in-water (O/W) emulsions, with or without avocado peel and seed extracts, during simulated gastric (**A**,**C**) and intestinal (**B**,**D**) digestion conditions. (**A**,**B**) = O/W dispersions; (**C**,**D**) = Emulsions containing LMP.

**Figure 4 foods-10-02193-f004:**
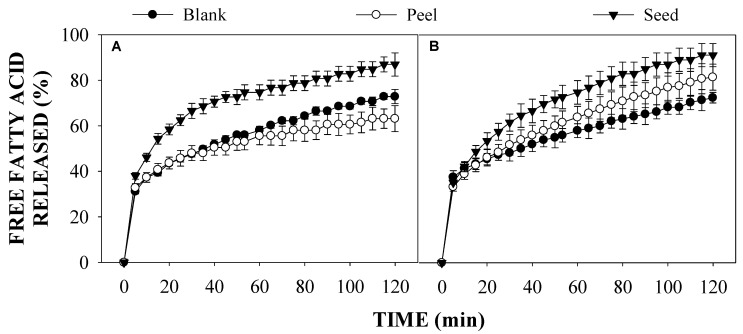
Effect of low methoxyl pectin (LMP) on the free fatty acid release in oil-in-water (O/W) emulsions containing avocado peel and seed extracts. (**A**) = O/W dispersions; (**B**) = Emulsions containing LMP.

**Figure 5 foods-10-02193-f005:**
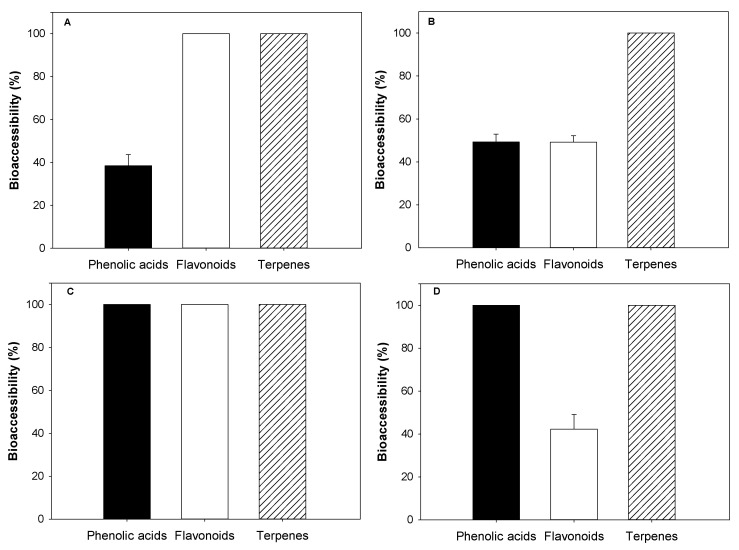
Bioaccessibility of phenolic acids, flavonoids and terpenes of avocado peel (**A**,**B**) and seed (**C**,**D**) extracts of oil-in-water (O/W) dispersions (**A**,**C**) and emulsions stabilized with LMP (**B**,**D**).

**Table 1 foods-10-02193-t001:** Elution gradients for identifying phenolic compounds by UPLC-MS.

Time (min)	Acetic Acid (%)	Acetonitrile (%)
0	95	5
5	90	10
10	87.6	12.4
18	72	28
23	0	100
25.5	0	100
27	95	5
30	95	5

**Table 2 foods-10-02193-t002:** Tentative quantification of phenolic compounds in the avocado peel and seed extracts by UPLC-ESI-MS/MS.

No.	Phenolic Compound	MW (g/mol)	SRM Quantification	Cone Voltage (V)	Collision Energy (eV)	Standard in Which Has Been (Tentatively) Quantified	Peel	Seed
	*Phenolic acids*						*1111.54 ± 11.25*	*377.98 ± 111.64*
1	p-hydroxybenzoic acid	138	137 > 93	30	15	p-Hydroxybenzoic Acid	0.39 ± 0.02	0.12 ± 0.01
2	Vanillin	152	151 > 136	25	10	Caffeic Acid	0.10 ± 0.02	0.02 ± 0.01
3	Vanillic acid	168	167 > 123	30	10	Caffeic Acid	0.03 ± 0.00	n.d.
4	Syringic acid	198	197 > 182	30	10	Caffeic Acid	0.09 ± 0.02	n.d.
5	Protocatechuic acid	154	153 > 109	40	15	Protocatehuic Acid	2.20 ± 0.24	0.28 ± 0.13
6	Protocatechuic acid glucoside	316	315 > 153	40	20	Protocatehuic Acid	26.1 ± 0.27	1.95 ± 0.51
7	Hydroxytyrosol	154	153 > 123	35	10	Hydroxytyrosol	0.19 ± 0.09	0.18 ± 0.06
8	Hydroxytyrosol glucoside	316	315 > 153	40	20	Hydroxytyrosol	0.14 ± 0.01	0.91 ± 0.23
9	Hydroxysalidroside	316	315 > 135	40	30	Hydroxytyrosol	0.10 ± 0.02	5.30 ± 1.08
10	Hydroxytyrosol glucoside arabinoside	448	447 > 153	40	20	Hydroxytyrosol	2.54 ± 0.11	n.d.
11	Tyrosol glucoside	300	299 > 137	40	20	Tyrosol	8.30 ± 0.52	10.40 ± 2.70
12	Salidroside	300	299 > 179	40	10	Tyrosol	0.54 ± 0.10	148.40 ± 50.30
13	Tyrosol glucoside arabinoside	432	431 > 137	40	20	Tyrosol	62.9 ± 1.09	0.75 ± 0.30
14	p-coumaric acid	164	163 > 119	35	10	p-Cumaric Acid	0.25 ± 0.01	0.20 ± 0.02
15	Coumaric acid glucoside	326	325 > 163	40	20	p-Cumaric Acid	0.33 ± 0.00	n.d.
16	Coumaroylquinic acid	338	337 > 191	40	20	p-Cumaric Acid	1.38 ± 0.06	3.61 ± 1.35
17	Caffeic acid	180	179 > 135	35	15	Caffeic Acid	0.42 ± 0.11	0.59 ± 0.14
18	Caffeic acid glucoside	342	341 > 179	40	20	Caffeic Acid	0.37 ± 0.06	0.09 ± 0.01
19	Caffeic acid glucoside derivative	546	545 > 341	40	20	Caffeic Acid	0.08 ± 0.01	n.d.
20	Dihydrocaffeic acid glucoside	344	343 > 181	40	20	Caffeic Acid	0.08 ± 0.00	0.21 ± 0.04
21	Caffeoylshikimic acid	336	335 > 161	40	20	Caffeic Acid	0.27 ± 0.01	2.21 ± 0.77
22	3-*O*-caffeoylquinic acid	354	353 > 179	40	15	5-O-Caffeoylquinic Acid	17.60 ± 0.26	176.20 ± 53.40
23	4-*O*-caffeoylquinic acid	354	353 > 173	40	15	5-O-Caffeoylquinic Acid	9.56 ± 0.07	10.70 ± 4.12
24	5-*O*-caffeoylquinic acid	354	353 > 191	40	15	5-O-Caffeoylquinic Acid	969.20 ± 9.21	11.30 ± 4.26
25	Dicaffeoylquinic acid	516	515 > 191	40	30	5-O-Caffeoylquinic Acid	1.05 ± 0.01	n.d.
26	Ferulic acid	194	193 > 134	30	15	Ferulic Acid	0.25 ± 0.01	0.06 ± 0.01
27	Ferulic acid glucoside	356	355 > 193	40	20	Ferulic Acid	4.14 ± 0.12	1.52 ± 0.56
28	Dihydroferulic acid glucoside	358	357 > 195	40	20	Ferulic Acid	0.24 ± 0.01	0.02 ± 0.00
29	4-*O*-feruoylquinic acid	368	367 > 173	40	20	Ferulic Acid	0.49 ± 0.05	0.22 ± 0.06
30	5-*O*-feruoylquinic acid	368	367 > 191	40	15	Ferulic Acid	1.61 ± 0.06	0.12 ± 0.6
31	3-*O*-feruoylquinic acid	368	367 > 193	40	15	Ferulic Acid	0.76 ± 0.03	2.83 ± 1.57
	*Flavonoids*						*5721.88 ± 51.73*	*1135.77 ± 456.57*
32	Catechin	290	289 > 245	40	15	Catechin	n.d.	280.50 ± 148.80
33	Epicatechin	290	289 > 245	40	15	Epicatechin	1891.00 ± 75.70	360.0 ± 140.60
34	Catechin glucoside	452	451 > 289	40	25	Catechin	2.40 ± 0.04	3.85 ± 0.08
35	Epicatechin glucoside	452	451 > 289	40	25	Epicatechin	7.89 ± 0.16	4.81 ± 0.66
36	Epigallocatechin	306	305 > 125	40	15	Epicatechin	6.27 ± 0.29	1.86 ± 0.67
37	Epicatechin gallate	442	441 > 169	40	20	Epicatechin	n.d.	1.39 ± 0.36
38	Catechin derivative	740	739 > 289	40	30	Catechin	1.55 ± 0.33	2.32 ± 0.27
39	Epicatechin derivative	740	739 > 289	40	30	Epicatechin	67.3 ± 9.96	1.72 ± 0.17
40	Dimer (type A)	576	575 > 285	40	20	Dimer B2	4.45 ± 0.09	6.28 ± 8.88
41	Dimer (type B)	578	577 > 289	40	20	Dimer B2	2262.0 ± 63.00	207.7 ± 112.10
42	Trimer (type A)	864	863 > 411	40	30	Dimer B2	9.27 ± 1.21	231.4 ± 74.20
43	Trimer (type B)	866	865 > 287	60	30	Dimer B2	383.2 ± 14.90	11.6 ± 16.50
44	Tetramer	1154	1153 > 865	70	20	Dimer B2	106.20 ± 4.18	9.73 ± 2.91
45	Pentamer	1442	1441 > 1028	80	25	Dimer B2	1.09 ± 0.06	0.55 ± 0.12
46	Hexamer	1730	1729 > 1153	80	30	Dimer B2	6.80 ± 0.02	1.31 ± 0.15
47	Quercetin	302	301 > 151	40	15	Quercetin-3-O-Glucoside	0.70 ± 0.02	0.39 ± 0.00
48	Quercetin arabinoside	434	433 > 300	40	20	Quercetin-3-O-Glucoside	0.70 ± 0.01	0.62 ± 0.03
49	Quercetin glucoside	464	463 > 300	40	30	Quercetin-3-O-Glucoside	20.30 ± 0.76	4.51 ± 0.11
50	Quercetin rhmanoside	478	477 > 301	40	25	Quercetin-3-O-Glucoside	2.07 ± 0.07	n.d.
51	Quercetin glucuronide	478	477 > 301	40	25	Quercetin-3-O-Glucoside	67.8 ± 0.18	0.04 ± 0.00
52	Quercetin acetylglucoside	506	505 > 300	40	25	Quercetin-3-O-Glucoside	2.99 ± 0.09	0.02 ± 0.02
53	Quercetin arabinoside glucoside	596	595 > 300	40	30	Quercetin-3-O-Glucoside	374.40 ± 22.70	0.39 ± 0.08
54	Quercetin rutinoside	610	609 > 300	40	30	Quercetin-3-O-rutinoside	6.73 ± 0.19	0.23 ± 0.11
55	Quercetin diglucoside	626	625 > 300	40	30	Quercetin-3-O-Glucoside	294.00 ± 13.1	1.82 ± 0.17
56	Quercetin glucoside rhamnoside	756	755 > 300	40	35	Quercetin-3-O-Glucoside	4.98 ± 1.12	n.d.
57	Isorhamnetin	316	315 > 300	40	15	Isorhamnetin	0.01 ± 0.00	n.d.
58	Isorhamnetin derivative	316	300 > 315	40	15	Isorhamnetin	2.19 ± 0.09	n.d.
59	Isorhamnetin arabinoside	448	447 > 315	40	20	Isorhamnetin	0.04 ± 0.00	0.49 ± 0.13
60	Isorhamnetin glucoside	478	477 > 315	40	20	Isorhamnetin	0.07 ± 0.00	0.05 ± 0.02
61	Isorhamnetin glucuronide	492	491 > 315	40	20	Isorhamnetin	9.20 ± 0.06	n.d.
62	Isorhamnetin arabinoside glucoside	610	609 > 315	40	30	Isorhamnetin	0.11 ± 0.01	n.d.
63	Kaempferol arabinoside	418	417 > 284	40	20	Kaempferol-3-O-Glucoside	0.26 ± 0.01	0.10 ± 0.03
64	Kaempferol glucoside	448	447 > 284	40	20	Kaempferol-3-O-Glucoside	4.81 ± 0.18	0.65 ± 0.01
65	Kaempferol rutinoside	594	593 > 284	40	30	Kaempferol-3-O-Glucoside	13.2 ± 0.27	n.d.
66	Kaempferol arabinoside glucoside	580	579 > 284	40	30	Kaempferol-3-O-Glucoside	160.7 ± 2.46	0.25 ± 0.03
67	Naringenin	272	271 > 151	40	15	Quercetin-3-O-Glucoside	0.27 ± 0.09	0.20 ± 0.01
68	Naringenin glucoside	434	433 > 271	40	20	Quercetin-3-O-Glucoside	0.59 ± 0.00	0.86 ± 0.14
69	Sakuratetin	286	285 > 199	40	20	Quercetin-3-O-Glucoside	0.25 ± 0.01	n.d.
70	Luteolin	286	285 > 133	40	20	Quercetin-3-O-Glucoside	n.d.	0.06 ± 0.02
71	Luteolin arabinoside glucoside	580	579 > 285	40	30	Quercetin-3-O-Glucoside	6.07 ± 0.34	n.d.
	*Terpenes*						*2.82 ± 1.86*	*0.62 ± 0.13*
72	Penstemide	444	443 > 119	40	25	Quercetin-3-O-Glucoside	2.82 ± 1.86	0.62 ± 0.13
	*Total phenolic compounds*						*6836.32 ± 64.66*	*1514.62 ± 578.33*

Data are expressed in mg per 100 g of extract as mean ± standard deviation (*n* = 3).

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
