# Peer review of "Lipid Digestibility and Polyphenols Bioaccessibility of Oil-in-Water Emulsions Containing Avocado Peel and Seed Extracts as Affected by the Presence of Low Methoxyl Pectin"

_foods, 2021, doi:10.3390/foods10092193_

Round 1

Reviewer 1 Report

The manuscript titled "Lipid digestibility and polyphenols bioaccessibility of oil-in-water emulsions containing avocado peel and seed extracts as affected by the presence of low methoxyl pectin" has been carefully reviewed, but there are too many writing flaws and errors that make it difficult to understand.

Line 4, low methoxyl pectin in the Title

Line 9, 13, 16, 19-21, LMP in the Abstract

Line 87, 91-97, LMP in the Introduction

Line 102, Low methoxyl pectin (LMP) in the Materials and Methods

Line 269, 286, low methoxyl pectin (LMP)

Line 123-124, AcQuity BEH C18 column, subscripts of C18 should not be allowed.

Table 2 and Table 3 should be able to be combined, and part of the analysis and mass spectrometry conditions can be added to supplementary materials or removed.

Mass spectrometry has a strong qualitative identification ability, but the quantitative stability is very poor. Abundance cannot be used as a quantitative benchmark for molecular identification. All data without quantitative standards are wrong and only suitable for semi-quantitative descriptions.

Author Response

Dear reviewer,

We would like to thank you for your contribution to our manuscripts. We have re-submitted a new version of our manuscript, highlighting the changes in red so you can easily notice them. After addressing each of your kind suggestions, we have strenghten our manuscript and enhanced its clarity and readiness. Please find attached to this response a point-by-point response to your comments.

Reviewer 2 Report

The manuscript entitled Lipid digestibility and polyphenols bioaccessibility of oil-in-water emulsions containing avocado peel and seed extracts as affected by the presence of low methoxyl pectin is a very interesting work which uses the avocado industrial residues in an efficient way to increase their impact on consumers’ health. The use of an emulsion is an effective strategy, however more stable formulations (such as microemulsions) would be more appropriate in order to study the effect of bioactive compounds without taking into account coalescence and other destabilization phenomena. The authors have used the appropriate bibliography while the figures and the tables are the most appropriate. The manuscript fits totally on the scope of the Special Issue Plant Bioactive Compounds in Foods and Food Packages. Overall, the study is an interesting, well organized but not an innovative work in terms of methods. If the authors would clarify the listed points, the manuscript could be accepted after major revision. Specifically, the following issues should be addressed in the revised manuscript.

Major Issues

  1. Please add a section regarding the stability of the formulations. DLS could be a useful tool in order to study the alteration of the droplets’ diameter versus time. The study should be conducted in the absence and in the presence of AP and AS extracts. The measurement of the electrical charge (ζ-potential) could also give some insights regarding the stability of the systems.
  2. Line 367. The distribution of the different compounds (phenolic, flavonoids) in the oil, aqueous phase and surfactant layer play a crucial role. The authors should find a way (NMR) in order to define which of the compounds may interact with the interface-surfactant layer. (See ref. Mitsou, E., Dupin, A., Sassi, A. H., Monteil, J., Sotiroudis, G. T., Leal-Calderon, F., & Xenakis, A. (2019). Hydroxytyrosol encapsulated in biocompatible water-in-oil microemulsions: How the structure affects in vitro absorption. Colloids and Surfaces B: Biointerfaces, 184, 110482.) The phenolic compounds primarily are hydrophilic however many of them have been described as amphiphilic compounds. Those compounds may interfere strongly with the LMP as the authors comment, however, more experimental data needed.
  3. The authors should clearly describe in the Introduction section why they decided to use LMP as surfactant in order to support the importance of the present study.

Minor Issues

  1. In the abstract please define the LMP surfactant abbreviation.
  2. Line 48. The authors should comment on the existence of the mucus layer in the gastrointestinal tract (intestinal epithelium). Mucus is a barrier which can strongly affect the bioavailability of substances where factors such as the charge and the lipophilicity among others, play a crucial role.
  3. Line 57. Please use amphiphilic molecule or surfactant, not the term amphiphilic surfactant.
  4. Line 60. Please also add the word safe.
  5. Please use the abbreviation Ο/W in the whole text as oil-in-water (O/W) has been used in line 68.
  6. In Fig. 1 the results which are presented refer to one system or the mean of many replications. Please define and add STDEV.

Author Response

Dear reviewer,

We would like to thank you for your kind comments and your valuable contribution to our manuscript. We have addressed both major and minor issues you have pointed out, and also have the answer to them as a point-by-point response. As a result, the main goal of our study is now properly depicted in the manuscript, and justified by an enhanced introduction section. We have re-submitted a new version of our manuscript, highlighting the changes in red so you can easily notice them. Therefore, after addressing each of your kind suggestions, we have strengthen our manuscript and enhanced its clarity and readiness. Please find attached to this response a point-by-point response to your comments.

Round 2

Reviewer 1 Report

This draft has been appropriately improved by the authors to be easy to read. However, there are still some structural flaws that need to be revised. The results of quantitative analysis with mass spectrometry are only suitable for comparison of differences between samples, and the applicability of food processing deserves a more in-depth discussion.

Author Response

Thank you for your comments and observations. Certainly, our manuscript has been enhanced and strengthen after the changes suggested. In this case, several statements highlighting that food processing may cause different results in the bioaccessibility of either avocado peel or seed extracts. In addition, we have suggested that further studies addressing the impact of food processing on the bioaccessibility of these PC should be performed. However, the main goal of our manuscript was to evaluate the digestibility of a food-grade encapsulation matrix, before its addition to a conventional or processed food product. Enclose to this reply you may find a detailed response about the changes included in our manuscript to highlight the impact of food processing on the bioaccessibility of PC.

Reviewer 2 Report

The manuscript has been improved based on the reviewers' comments. 

Author Response

Thank you for your kind comments and insightful observations. Certainly, our manuscript has been enhanced and strengthen after the changes suggested.